# DELM: A PYTHON TOOLKIT FOR DATA EXTRACTION WITH LANGUAGE MODELS

## ABSTRACT

Large Language Models (LLMs) have become powerful tools for annotating unstructured data. However, most existing workflows rely on ad hoc scripts, making reproducibility, robustness, and systematic evaluation difficult. To address these challenges, we introduce DELM (Data Extraction with Language Models), an open-source Python toolkit designed for rapid experimental iteration of LLM-based data extraction pipelines and for quantifying the trade-offs between them. DELM minimizes boilerplate code and offers a modular framework with structured outputs, built-in validation, flexible data-loading and scoring strategies, and efficient batch processing. It also includes robust support for working with LLM APIs, featuring retry logic, result caching, detailed cost tracking, and comprehensive configuration management. We showcase DELM's capabilities through two case studies: one featuring a novel prompt optimization algorithm, and another illustrating how DELM quantifies trade-offs between cost and coverage when selecting keywords to decide which paragraphs to pass to an LLM.

## 1 INTRODUCTION

Transforming data into structured format has long been a challenge in NLP. Early solutions relied on handcrafted regular expressions and rule-based systems; later, supervised models for named entity recognition (NER) and relation extraction (RE) became dominant. All of these approaches require either painstaking rule writing or sizable labeled datasets. Large language models (LLMs) now offer a compelling alternative: a user can describe the task in natural language and obtain structured outputs with little or no training. Techniques such as few-shot prompting (Brown et al., 2020), fine-tuning (Wei et al., 2022), and prompt engineering (Schulhoff et al., 2025) steer models toward high-quality extractions, while constrained decoding and token masking can enforce structured formats (Willard & Louf, 2023). These advances have enabled applications ranging from financial-report parsing to clinical-note mining without building task-specific models from scratch.

However, building reliable LLM-based extraction pipelines remains difficult in practice. LLM outputs can be highly sensitive to prompts and alignment techniques, and API behavior varies across providers. Practitioners therefore fall back on brittle scripts and manual prompt tweaks, sacrificing reproducibility and hindering rigorous comparison across experiments. Scaling exacerbates the problem: one must batch requests, handle rate limits and transient errors, cache results to avoid redundant billing, monitor token usage, and log every interaction for auditability. Without an integrated framework, lost intermediate results, undetected configuration drift, and mounting costs are commonplace. All these issues underscore the need for a more disciplined approach to LLM-based data extraction.

We introduce DELM (Data Extraction with Language Models), an open-source toolkit that brings structure to LLM-based data extraction while maintaining the flexibility required for research. DELM provides a unified, modular workflow: configurable loaders can ingest data from a variety of formats, preprocessing hooks prepare inputs, and a batched executor interfaces with LLM APIs. Extraction quality is assessed using interchangeable scoring modules. The toolkit also includes robust engineering safeguards, such as automatic retries with exponential backoff, deterministic caching keyed to the prompt and model, and fine-grained cost tracking, all of which help avoid wasted tokens and lost progress. Crucially, DELM enables reproducible experimentation by organizing each run

around a declarative configuration file that records data provenance, prompt details, model parameters, and evaluation metrics.

We demonstrate the value of this framework through two empirical case studies. In the first, we analyze the trade-off between monetary cost and extraction coverage when using keyword-based filtering to select which paragraphs are passed to the LLM. In the second, we integrate a prompt-optimization loop within DELM to show how systematic exploration improves extraction performance while DELM seamlessly manages logging, caching, and cost monitoring.

By decoupling scientific inquiry from engineering overhead, DELM empowers researchers to concentrate on high-level methodological questions without repeatedly "reinventing the wheel" of pipeline engineering. Ultimately, our toolkit aims to accelerate the development of reliable LLM-based extraction methods and encourage more rigorous, reproducible evaluations.

## 2 RELATED WORK

### 2.1 LLMS FOR INFORMATION EXTRACTION

LLMs are rapidly transforming the landscape of information extraction (IE). Early work demonstrated that prompting or lightweight fine-tuning could unify subtasks such as NER and RE into a single generative framework. (Xu et al., 2024) provide a comprehensive survey of this transition, reframing IE as a sequence generation problem.

Further, (Willard & Louf, 2023) demonstrate an efficient constrained generation algorithm for restricting model outputs to valid JSONs. Some LLM APIs, including all frontier labs'[1], allow developers to define a JSON response schema and reliably get outputs in that format. The users typically have to specify the schema with Pydantic classes, but the specifc way to do it varies by provider. Instructor, a Python library, introduced by (Liu & contributors, 2025) handles most major LLM APIs.

Empirical studies in domain-specific settings confirm the practical value of these approaches. For example, (Dagdelen et al., 2024) fine-tune GPT-3 and LLaMA-2 on 100-500 annotated passages to jointly extract materials science entities and relations, achieving strong performance in an ultra-low data regime. Their best-performing model outperforms prior baselines such as MatBERT-Proximity and Seq2rel. They further demonstrate that incorporating a human-in-the-loop component yields substantial efficiency gains: once an intermediate model is trained on just 300 examples, annotation time per abstract drops by 57%, and per-token effort by nearly 60%.

At the same time, prompt design remains a critical component of LLM-based IE. (Schulhoff et al., 2025) catalog 58 prompting strategies and introduce a taxonomy that covers zero-shot, few-shot, thought-generation, ensembling, self-criticism, and decomposition. They also discuss alignment challenges covering ambiguity, biases, calibration, and prompt-sensitivity. They also present a case study showing vide variation in scores depending on the prompting method.

Together, these studies point to the trend for replacement of task-specific architectures with unified LLM-based approaches. This shift, enabled by recent advances in LLM capabilities, offers robust performance gains across diverse IE tasks. However, these gains are contingent on effective design of these systems. As a result, there is a growing need for frameworks that facilitate systematic experimentation, reproducibility, and efficient deployment of LLMs for IE.

### 2.2 EXISTING SOFTWARE

A number of open-source libraries have emerged to support LLM-based IE pipelines, though they vary in focus and scope. LangChain provides a modular programming framework for composing LLM applications as chains of prompt templates, retrievers, memory modules, and output parsers (Harrison & contributors, 2025). Its primary goal is production-oriented orchestration, however, LangChain offers limited built-in support for systematic evaluation, ablation, or experiment tracking. By contrast, DSPy introduces a declarative abstraction over LLM pipelines (Khattab et al., 2023). Each step in a DSPy workflow is expressed as a transformation operator, and the system includes a

---

[1]OpenAI, Anthropic, Google AI, xAI

compiler that automatically optimizes prompt templates, hyperparameters, and in-context examples to maximize performance on user-specified metrics. DSPy thus treats pipeline construction as a learnable and optimizable process, enabling structured exploration of prompt and demonstration design.

While LangChain and DSPy offer powerful abstractions for building LLM applications and optimizing prompt pipelines, they are not primarily focused on experimental workflows for IE. DELM complements these tools by providing a configuration-driven framework tailored to empirical IE research, with features for schema validation, batch execution, caching, and systematic evaluation.

## 3 DESIGN PRINCIPLES

The design of DELM is informed by three principles: (i) framing data extraction as a predictive task; (ii) engineering for scalability and fault tolerance; and (iii) ensuring reproducibility and provenance.

### 3.1 DATA EXTRACTION AS A PREDICTIVE TASK

A central principle underlying DELM is to frame LLM-based data extraction explicitly as a predictive modeling task. By conceptualizing extractions as predictions, researchers can quantify the performance of their pipelines in a comparable way. For example, extraction of binary or categorical variables can be systematically evaluated using standard classification metrics such as accuracy, precision, recall, and F-score, whereas continuous variable extraction can leverage regression metrics like $R^2$ or mean absolute error (MAE). To facilitate this, DELM incorporates interchangeable scoring modules, providing users the flexibility to measure and compare the performance of different extraction approaches in a structured manner.

A key implication of this predictive framing is the importance of dataset representativeness, often formalized via the assumption of independent and identically distributed (i.i.d.) observations. While DELM does not enforce i.i.d. constraints, evaluations based on representative samples are essential for obtaining meaningful performance metrics. Just as supervised models evaluated on biased data yield unreliable estimates, LLM extraction pipelines assessed on skewed samples will produce distorted results. DELM simplifies the measurement of such effects across prompting and fine-tuning modes.

### 3.2 SCALABILITY

Scaling from small-scale tests to large-scale production introduces engineering challenges that could compromise results and inflate costs. Issues such as transient API failures, network interruptions, and rate limits become inevitable at scale. A scalable pipeline must anticipate and handle these errors gracefully. Ad hoc scripts often lack this foresight, leading to aborted runs and wasted resources. DELM systematically addresses these challenges by implementing automated retry logic with exponential backoff, reducing the likelihood of job termination due to temporary errors. Additionally, DELM mitigates the risk of data loss from interruptions by persisting intermediate results to durable storage. It also implements caching to minimize redundant computations and lower costs associated with repeated executions, particularly beneficial when similar pipelines undergo testing.

### 3.3 REPRODUCIBILITY AND PROVENANCE

Reliable empirical work requires that extraction results be fully attributable to the data, prompts, model versions, and configuration choices that produced them. Small, undocumented changes in any of these inputs can shift apparent performance and confound comparisons across experiments. We therefore treat configuration as data: every run records data sources, sampling filters, prompt text, model and temperature settings, and evaluation parameters. Deterministic caching keyed to these artifacts both prevents inadvertent recomputation and guarantees that identical configurations reproduce identical results. Persistent logs (including raw model responses and validation outcomes) provide an auditable trail. DELM operationalizes this provenance-first design so that experiments remain comparable over time and across settings.

## 4 ARCHITECTURE

### 4.1 SYSTEM OVERVIEW

DELM structures extraction as a configurable pipeline that separates concerns between data handling, model interaction, validation, and evaluation. The end-to-end flow begins with data loading and preprocessing, proceeds through optional relevance scoring and filtering, executes batched LLM calls with structured outputs, performs cache operations, conducts alignment and schema validation, and concludes with logging and optional persistence of results. Throughout this process, the system maintains comprehensive provenance records and cost tracking while supporting interruption-safe resumption.

The pipeline operates through distinct phases. Loaders first ingest heterogeneous inputs spanning text files, tabular data (CSV, Parquet), PDFs, Word documents, and HTML to produce a uniform tabular representation. Preprocessing then segments long records into manageable chunks according to user-specified rules. When enabled, a relevance scorer filters chunks to optimize the trade-off between extraction coverage and computational cost. The batched executor subsequently invokes the LLM through Instructor introduced by (Liu & contributors, 2025) with a predefined Pydantic schema inroduced by (Colvin et al., 2025), leveraging concurrent execution, retry logic, and deterministic caching keyed to the fully rendered prompt, schema, model, and generation parameters. The system validates returned objects against the user-defined schema before persisting results to disk and creating audit logs.

### 4.2 CORE COMPONENTS

The implementation maintains clean separation between orchestration logic, pluggable behaviors, and shared utilities through four primary modules. The core module handles pipeline orchestration and persistence through three key components: the Data Processor manages loading, chunking, and optional scoring and filtering operations; the Extraction Manager coordinates batched Instructor calls, concurrent execution, cache integration, and schema-aligned parsing; and the Experiment Manager oversees run directories, checkpointing, state restoration, and output consolidation.

The schemas module encompasses schema loading, rendering, and validation functionality. The Schema Manager loads YAML specifications, constructs corresponding Pydantic models, and renders prompts dynamically. During inference, the system validates all responses against these same schemas to ensure structured, comparable outputs across experiments.

The strategies module provides pluggable behaviors across multiple dimensions. Loaders accommodate diverse input formats including plain text, tabular data (CSV, Parquet, Feather, Excel), PDFs processed through marker OCR, Word documents, and HTML or Markdown files. Scoring strategies encompass both keyword-based and fuzzy matching variants for relevance assessment. Splitting strategies govern how the system constructs chunks from longer documents.

The utils module implements engineering safeguards and shared infrastructure components. This includes the Retry Handler with exponential backoff, a Semantic Cache using SQLite WAL by default with LMDB and file system alternatives, a Cost Tracker providing token counting and cost estimation, logging configuration utilities, concurrent processing helpers, and provider pricing metadata.

### 4.3 ROBUSTNESS

Scaling extraction to production environments demands defenses against transient failures, configuration drift, and computational waste. DELM addresses these challenges through five integrated mechanisms. The system implements automatic retries with exponential backoff around all model calls to handle API rate limits and temporary failures. A persistent, deterministic semantic cache keys API responses to the fully materialized prompt, system prompt, model identifier, and generation parameters, eliminating redundant computations across experiments, saving time and money whenever possible.

Fine-grained token accounting and cost estimation with optional budget constraints prevent unexpected expenses during large-scale extractions. Checkpointed batching and resumability features

protect against lost progress due to interruptions or failures. Centralized, configurable logging creates comprehensive audit trails for debugging and reproducibility analysis.

The executor queries the cache before each API call and inserts validated results following cache misses. Retry mechanisms implement exponential backoff to respect API rate limits while maintaining throughput. Cost tracking meters both input tokens and structured output tokens separately. Every experimental run persists its complete configuration and schema specifications to disk, enabling exact reproduction and facilitating post hoc analysis of extraction performance across different parameter settings.

## 5 CASE STUDIES

### 5.1 COST-VS-RECALL TRADEOFF IN KEYWORD FILTERING

When performing LLM-based information extraction, passing every paragraph potentially maximizes recall but could be prohibitively expensive. We demonstrate how this cost–recall tradeoff could be quantified using DELM's configurable preprocessing and evaluation stack and data from (Fukui et al., 2024).

The task involves extracting mentions of commodities (their types, prices, units, etc) from investor call transcripts. We split the records 80/20 into train and test sets. We use a predefined set of keywords to filter paragraph. We then apply a greedy forward selection strategy: at step $k \in \{1, \ldots, N\}$, we add the keyword that maximizes train recall on `commodity_prices.good`, holding previously chosen keywords fixed. This produces nested keyword sets of size $k = 1, \ldots, N$. For each configuration, we run DELM's performance estimator to compute recall and DELM's cost estimator to obtain an estimated total cost, scaling from sampled API calls to the full split. Within each split, costs are normalized as

$$\tilde{C}_k = \frac{C_k}{\max_j C_j} \in [0, 1],$$

where $\max_j C_j$ denotes the maximum estimated cost across all evaluated keyword sets for that split. Figure 1 reports the resulting Pareto frontier of normalized cost versus recall, together with a constrained quadratic fit ($a \le 0$) summarizing the trend.

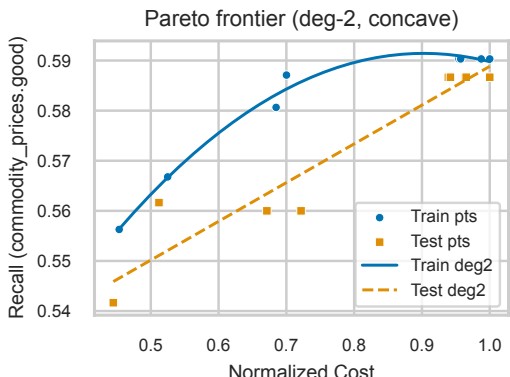

Figure 1: *Cost-recall Pareto frontier for the extraction task.* X-axis: Normalized cost, computed as the estimated total cost divided by the maximum estimated total cost across all evaluated keyword configurations within each split (Train/Test), with normalization done independently per split. Y-axis: Recall for commodity type. Points denote greedy keyword sets ($k = 1..N$) on Train and Test. Curves show degree-2 polynomial fits with the quadratic term constrained to be non-positive for Train and Test.

### 5.2 PROMPT OPTIMIZATION

The next example showcases the use of DELM for iterative prompt optimization. There is growing literature on automatic prompt optimization (e.g. (Ramnath et al., 2025)), framing the prompt search as an optimization problem over discrete or continuous prompt spaces.

The problem can be formalized as follows. Let $\mathcal{D} = \{(x_i, y_i)\}_{i=1}^{N}$ be texts $x_i$ with reference labels $y_i$ for a target field. Given a prompt $p \in \mathcal{P}$, the extractor (LLM) $f_p : \mathcal{X} \to \widehat{\mathcal{Y}}$ produces $\hat{y} = f_p(x)$. Let $\phi : \mathcal{Y} \times \widehat{\mathcal{Y}} \to [0, 1]$ be a correctness indicator (e.g., precision). The objective is

$$p^{\star} \in \arg\max_{p \in \mathcal{P}} J(p) \quad \text{with} \quad J(p) = \mathbb{E}_{(x,y)\sim\mathcal{D}}\big[\phi\big(y, f_p(x)\big)\big].$$

We optimize $p$ iteratively by sampling mini-batches, evaluating errors, and updating $p$ using an optimizer LLM $g$ with a meta-prompt.

> **Meta-prompt example**
>
> Review the examples below. Each example is taken from paragraphs where $price\_expectation$ is identified incorrectly. $price\_expectation$ is a binary variable. Your task is to extract and articulate the underlying logic or reasoning that justifies why these examples are misclassified. Then, incorporate this implicit logic into the prompt itself as explicit guidance. Important: Do not remove or alter any existing information in the prompt, only add clarifying logic derived from the examples.
> Examples: {examples}
> Prompt: {prompt}

---

**Algorithm 1** LLM-In-the-Loop PRompt Optimization (LILPRO)

---

**Require:** Dataset $\mathcal{D} = \{(x_i, y_i)\}_{i=1}^{N}$, extractor $f$, initial prompt $p_0$, batch size $B$, batches $T$, error budget $k$

1: **for** $t = 0, 1, \ldots, T-1$ **do**
2:      $S_t \subset \{1, \ldots, N\}$ with $|S_t| = B$                 ▷ sample without replacement
3:      **for each** $i \in S_t$ **do**
4:          $\hat{y}_i \leftarrow f_{p_t}(x_i)$
5:      $m_t \leftarrow \frac{1}{B} \sum_{i \in S_t} \phi(y_i, \hat{y}_i)$             ▷ batch score (e.g., precision surrogate)
6:      $E_t \leftarrow \{i \in S_t : \phi(y_i, \hat{y}_i) = 0\}$                 ▷ error set
7:      **if** $E_t \neq \varnothing$ **then**
8:          $K_t \subseteq E_t$ with $|K_t| = \min(k, |E_t|)$      ▷ subsample (e.g., uniform) for curation
9:          $\mathcal{C}_t \leftarrow \{(x_i, y_i, \hat{y}_i) : i \in K_t\}$             ▷ curated error triplets
10:        $p_{t+1} \leftarrow g(p_t, \mathcal{C}_t)$            ▷ optimizer LLM proposes refined prompt
11:      **else**
12:          $p_{t+1} \leftarrow p_t$
13: **return** $p_T$                             ▷ optionally select $\arg\max_t m_t$

---

We run this algorithm using data from (Fukui et al., 2024) and the $price\_expectation$ variable and the results can be seen in Figure 2. We focus on this variable because the baseline definition of "expectations" is inherently open to interpretation. For instance, managers on these calls may discuss alternative growth scenarios conditional on commodity price levels, yet not all such statements qualify as expectations in the sense of a weighted average over possible outcomes. Thus, what counts as an expectation is arbitrary.

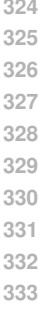

Figure 2: *Precision vs batch for the prompt-optimization experiment (presence).* X-axis: Batch index. Y-axis: Precision. Precision is computed using the estimator's presence-based metric for `commodityprices.price_expectation`, which credits a record when the correct boolean value appears at least once (duplicates do not affect the score).

## 6 CONCLUSION

We introduce DELM, a lightweight, configuration-driven toolkit for building, evaluating, and scaling LLM-based data-extraction pipelines. The framework centers on reproducibility via provenance tracking and deterministic caching, operational reliability through robust batching and retries, and rigorous measurement with schema-aware validation and cost accounting. Case studies demonstrate practical gains in the ease of implementation of information extraction tasks such as estimating cost–recall trade-offs and LLM-in-the-loop prompt optimization. Future directions for development include built-in time-based rate limits and agentic AI integrations.

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
