# OpenReview forum: "DELM: a Python toolkit for Data Extraction with Language Models"
_ICLR.cc/2026/Conference — ICLR 2026 Conference Withdrawn Submission_

### Official Review · Reviewer_c5fx · 2025-10-16

**Soundness:** 1
**Presentation:** 2
**Contribution:** 1
**Rating:** 2
**Confidence:** 4

**Summary:**

The paper introduces DELM, a Python toolkit designed to streamline LLM-based data extraction pipelines. The authors argue that existing workflows, often reliant on ad hoc scripts, suffer from poor reproducibility, robustness, and evaluability. DELM aims to address these issues by providing a configuration-driven, modular framework that incorporates features such as structured output validation, automatic API retries, result caching, and cost tracking. The paper presents two case studies to demonstrate the toolkit's capabilities: one analyzing the cost-recall tradeoff in a keyword-based filtering task, and another implementing a novel LLM-in-the-loop prompt optimization algorithm.

**Strengths:**

- **Addresses a Practical and Important Problem**: The paper correctly identifies a significant pain point in contemporary applied ML research: the difficulty of building reproducible, robust, and efficient LLM-based data processing workflows. A tool that solves this has high practical value for the community.
- **Sound Engineering Design Principles**: The framework is built on solid design principles, such as configuration-as-code for reproducibility, modularity for flexibility, and built-in robustness features. This demonstrates a thoughtful approach to system design for research, and two case studies demonstrate its effectiveness.

**Weaknesses:**

**1. Critical Lack of System-Level Evaluation**: The paper makes strong claims about scalability, robustness, and efficiency but provides no quantitative evidence. A systems-oriented paper must include experiments to measure its own performance. Missing experiments may include:

   - ***Scalability tests***: Throughput and resource utilization (API/memory) benchmarks on datasets of increasing size.
   - ***Robustness validation***: Ablation stuies showing the impact of the retry mechanism on task completion rate under simulated network failures.
   - ***Efficiency analysis***: Quantitative analysis of the caching feature, reporting cache hit rates and corresponding cost/time savings.

**2. Limited and Unconvincing Validation of Generality**: The paper claims DELM is a general-purpose toolkit but validates it with two very specific, bespoke case studies. This approach showcases what can be done but fails to prove the framework's utility and superiority for more common, standardized tasks. A more compelling validation would involve applying DELM to several public IE benchmarks and demonstrating a significant improvement compared to baseline implementations (e.g., LangChain or custom scripts).

**3. Insufficient Novelty**: The paper reads more like a tutorial or technical report for a system rather than a research paper with technical innovations. The main contribution is the integration of well-known engineering techniques into a domain-specific toolkit. This is a valuable engineering effort but lacks the academic novelty expected at ICLR. When compared to existing frameworks that tackle programmatic data processing or LLM pipeline optimization (e.g., Data-Juicer, DataFlow, DSPy), the proposed framework does not appear to offer a sufficiently novel paradigm or capability.

**4. Overstated Conceptual Contribution**: The principle of "Data Extraction as a Predictive Task" (Section 3.1) is presented as a cornerstone of the design. However, its implementation is simply the inclusion of a module for calculating standard evaluation metrics. This is a common practice in any evaluation harness, and framing it as a novel conceptual principle overstates its contribution. The framework does not seem to contain any unique mechanisms that are directly enabled by this "predictive framing" beyond what is trivial.

**Questions:**

Following up on the points raised in the Weaknesses:

1. Could the authors provide quantitative, empirical evidence to support these claims? Specifically, what is the system's throughput, how does it perform on large-scale datasets, and what are the measurable benefits of the caching and retry mechanisms?

2. The LILPRO algorithm is an interesting contribution. How does it compare, both conceptually and empirically, to existing automatic prompt optimization methods? A direct comparison would help situate its novelty and effectiveness.

3. Have the authors considered applying DELM to reproduce results from a published LLM-based IE study on a standard benchmark? And what unique advantages and features does the author think DELM has compared to other data processing frameworks (e.g. Data-Juicer and DataFlow)?

---

### Official Review · Reviewer_5uft · 2025-10-28

**Soundness:** 1
**Presentation:** 1
**Contribution:** 1
**Rating:** 2
**Confidence:** 4

**Summary:**

This paper introduces DELM, a Python package for data extraction using LLMs. The paper argues that DELMs solves several flaws in existing tools by implementing efficient retry logic, result caching, cost tracking, etc.. While I recognize that DELM might be a good package, there are inherent challenges in describing software in a paper. This paper does not provide sufficient information for me to know. Given the importance of the code in this paper, it saddens me that an attempt wasn’t made to share the code, e.g., using an anonymous GitHub or a similar service.


I recognize the need for scientific tooling and packages for LLM IE. I also acknowledge that high-quality software should be recognized; however, given the current state of the paper, I am unable to evaluate the quality of the work.

**Strengths:**

- This work clearly recognizes the need to build high-quality software and denotes several limitations in existing tools.
- From the description of functionality, I suspect they have produced a high-quality package to solve these. However, the current state of the paper does not allow me to determine this.

**Weaknesses:**

Besides the previously mentioned weaknesses:

- The introduction of existing software lacks a sufficient introduction of the plethora of relevant competitors, e.g., PydanticAI. I think it might also be worth discussing if a coherent package for LLM IE is needed, compared to combining a few existing key packages, such as outlines for managing structured generation and existing packages for managing API requests. This might be the case, however, I am unsure.

- It is argued (3.2) that scalable pipelines must handle API failures, network interruptions, and rate limits. etc.. However, it is not clear to me that DELM solves these problems better than existing tools, such as an exponential backoff decorator.

- Central statements such as “minimizes boilerplate code” and “offers a modular framework” are undocumented, and while I understand that these might be subjective, no code snippets are shared to allow subjective assessment. Given that the authors had plenty of space left, I would have loved to have seen a more convincing argument.

Minor:
- I believe the in-text citations are misformatted: "(Schulhoff et al., 2025) catalog 58 prompting strategies" should be "Schulhoff et al., (2025) catalog 58 prompting strategies"

**Questions:**

Can you provide comparable code snippets from your package and alternative approaches that demonstrate the relative benefits of the package?

---

### Official Review · Reviewer_rMzF · 2025-10-30

**Soundness:** 1
**Presentation:** 1
**Contribution:** 1
**Rating:** 2
**Confidence:** 3

**Summary:**

The authors present a library designed to facilitate the use of LLMs in scientific experiments. They provide two examples to demonstrate the library's application and capabilities.

**Strengths:**

The authors address an important challenge in maintaining scientific rigor when incorporating LLMs into experimental workflows. This focus on reproducibility and systematic evaluation when using LLMs in research has value to the community.

**Weaknesses:**

- The paper lacks definitions of key terminology. For instance, "information extraction" appears throughout the manuscript without context or a clear definition of scope, despite being a broad term that encompasses multiple NLP tasks. This ambiguity makes it difficult to assess the work's contributions.
- The manuscript provides insufficient detail for readers to understand the library's architecture independently of the code. An overview of the software architecture and design principles would help readers identify which design elements represent novel contributions beyond standard engineering practices like caching or exponential backoff. Furthermore, the relationship between the examples and core library functionality remains unclear. Without this information, it is difficult for the reader to understand the libraries APIs and abstractions. A key information, since the library appears to be the main contribution.
- The authors do not motivate “passing paragraphs to LLMs for extracting commodities from investor call transcripts” within a scientific context. The authors also don’t motivate keyword-based preprocessing when alternative retrieval methods such as regex or embeddings might be alternative options depending on the problem.
- In the prompt optimization section, the boundary between existing techniques from the literature and the authors' novel contributions needs clarification. While LILPRO appears to be a novel method for prompt optimization, this requires explicit confirmation and clearer delineation from prior work.

**Questions:**

- Could the authors more clearly scope and define what they mean by information extraction in the context of this work?
- Given the novelty of the prompt optimization method, have the authors conducted ablation studies or comparative evaluations against baseline prompting strategies?

---

### Official Review · Reviewer_M6CJ · 2025-10-31

**Soundness:** 2
**Presentation:** 2
**Contribution:** 2
**Rating:** 2
**Confidence:** 4

**Summary:**

This paper introduces DELM, an open-source Python toolkit designed to standardize and accelerate experimental research in LLM-based data extraction. The toolkit's goal is to address common issues like poor reproducibility and engineering overhead by providing a modular, configuration-driven framework. DELM's architecture is based on principles of treating extraction as a predictive task, ensuring scalability and fault tolerance, and prioritizing reproducibility through declarative configurations and deterministic caching. The paper demonstrates the toolkit's utility through two case studies: one analyzing the cost-recall trade-off of a keyword-based filtering strategy, and another implementing a novel LLM-in-the-loop prompt optimization algorithm called LILPRO.

**Strengths:**

- The design principles of DELM are sound and well-articulated. Framing extraction as a predictive task, building in robustness features (retries, checkpointing), and enforcing reproducibility via configuration management and caching are all excellent goals for such a framework.
- The proposed architecture appears modular and thoughtful, building upon established tools like Pydantic and Instructor to handle structured data and model interaction. This practical approach increases the likelihood of adoption.

**Weaknesses:**

- The paper's claims of scalability and suitability for large-scale experiments are entirely unsubstantiated. For a systems-focused paper, the absence of any performance benchmarks is a major flaw. There is no analysis of the framework's overhead, its throughput under concurrent loads, or the potential bottlenecks of its components (e.g., the default SQLite cache).
- The case studies, which serve as the primary evidence for the toolkit's utility, suffer from a critical lack of detail that prevents reproducibility. The LILPRO algorithm is missing its core components (the optimizer LLM model and the meta-prompt). The cost-recall study omits details about the initial keyword set. This makes it impossible for a reader to verify the results or build upon them, which undermines the paper's central theme of reproducibility.
- The evaluation methodology within the case studies is questionable. The LILPRO experiment relies on a lenient, non-standard presence-based precision metric without providing a comparison to standard, stricter metrics (e.g., exact match F1-score). This choice may present an overly optimistic picture of the prompt optimization's effectiveness.
- A significant methodological limitation is the handling of model stochasticity. The paper states that the cache is keyed to the configuration, including temperature. The Q&A reveals this means the first stochastic output is cached and reused. While this ensures run-level reproducibility, it completely obscures the inherent variance of LLM outputs, a critical factor in many research settings. A tool for scientific inquiry should facilitate the study of this variance, not hide it. This limitation is not discussed by the authors.

**Questions:**

- Could you provide performance benchmarks for DELM? Specifically, what is the throughput of the system (e.g., requests per second) under concurrent execution, and how does the performance of the default SQLite cache degrade as the number of parallel workers increases?
- The LILPRO case study is a central piece of evidence. To make it reproducible, could you please specify the exact optimizer LLM (`g`) used and provide the full meta-prompt that instructs it to refine the task prompt?
- In the LILPRO evaluation, why was a lenient presence-based metric used exclusively? Could you also report the results using standard, strict-match precision and F1-score to provide a more complete assessment of the performance improvement?

---

### Official Review · Reviewer_3fmM · 2025-11-01

**Soundness:** 3
**Presentation:** 2
**Contribution:** 2
**Rating:** 2
**Confidence:** 4

**Summary:**

The paper introduces DELM, an open-source Python toolkit for LLM-based information extraction (IE). It provides modular data loading and splitting, Pydantic-based schemas and validation, batched execution with retries, deterministic caching, cost tracking, and provenance-first experiment management. Two case studies illustrate use: a cost–recall analysis under keyword filtering and an LLM-in-the-loop prompt-optimization routine. The problem is timely given the shift toward generative IE with LLMs.

**Strengths:**

•	Solid engineering and provenance. The pipeline is modular with a clear separation between task logic, data handling, and configuration. Schema‑based validation catches malformed outputs early. Batching with backoff retries improves reliability. A deterministic cache keyed by prompt, schema, model, and parameters reduces cost and supports reproducibility. Token and cost tracking are first‑class features. Provenance is recorded so runs can be diffed, resumed, and audited.
•	Case studies demonstrate practical relevance. The cost–recall analysis with a keyword prefilter shows how to control budget while keeping coverage. The human in the loop prompt refinement example illustrates a reasonable workflow for iterative improvement with visible trade offs.
•	The writing is clear and accessible. Sections follow a coherent logic. Key terms are defined on first mention, and figures/tables are labeled clearly, which reduces barriers to understanding across technical backgrounds.

**Weaknesses:**

•	Algorithmic novelty and evidence are limited. LILPRO is a heuristic loop that samples batches, curates errors, and asks an optimizer LLM to rewrite the prompt. No new theory, convergence analysis, or controlled comparisons to existing automatic prompt-optimization methods are provided; only a precision-vs-batch curve is shown, without variance or significance.
•	Related work is underdeveloped and mostly qualitative. The sections summarize IE with LLMs and list LangChain/DSPy, but do not quantify differences or position DELM on shared tasks or metrics.
•	Missing system-level baselines and benchmarks. No side-by-side system baselines (LangChain/DSPy or a plain Python baseline) under identical tasks, models, and rate limits, and no reporting of throughput, failure rate, unit cost, wall-clock time, or variance/significance.
•	Ablations and component value: No plug and play ablations for cache, retry, budget limits, logging, or schema enforcement. There is no clear marginal utility of each component.
•	Missing architecture figure. Section 4 describes components and robustness, but there is no end-to-end architecture diagram that shows data/control flow, cache keys, backoff policy, schema-check locations, observability, and rate-limit handling. This hurts auditability.
•	Open-source and reproducibility are emphasized, however no anonymous artifacts (repo/zip with code, configs, prompts, and run scripts) are provided in the submission package.

**Questions:**

1.	Related Work coverage. It may help to expand Section 2.2 beyond LangChain and DSPy. Consider adding several IE toolkits or schema-constrained generation libraries, such as LMQL (token-level constrained decoding), OpenAI Structured Outputs (JSON Schema adherence), vLLM structured decoding (XGrammar/Guidance backends), or JSONformer (schema-driven constrained decoding), and stating the inclusion criteria so Section 2.1 remains conceptual while Section 2.2 lists concrete software.
2.	Baselines. Is it possible to add a direct comparison to DSPy on the same tasks, models, and rate limits and report throughput, failure rate, unit cost, and latency, with scripts to reproduce? A small table plus a short paragraph is enough. Also report p50/p95/p99 latency, failure breakdowns by 4xx/5xx/parse/schema, token/USD cost, and uncertainty (mean±SD, 95% CIs) under identical rate-limit settings.
3.	Ablations. It will be good to understand which component reduces failures the most. Show cache/retry/schema/budget/logging toggles. Include stress tests within provider-published rate limits, with long-context runs. Specify queueing and backoff policy and report p50/p95/p99 latency, 4xx/5xx/parse failure breakdown, and unit cost in tokens and USD, with mean±SD over at least three independent runs and 95% CIs. If you enforce schemas at generation time, clarify how this interacts with constrained decoding.
4.	Add an architecture figure. Include components, data/control flow, cache key and eviction policy, retry/backoff logic, schema enforcement points, observability (logs/metrics), and rate-limit behavior.
5.	Artifacts. Please provide an anonymous reproducibility bundle (repo or zip) with code, pinned dependencies, configs, prompts, Pydantic schemas, random seeds, a small public data slice, and one-command scripts to reproduce tables and figures.

---

### Note · Authors · 2025-11-24

I have read and agree with the venue's withdrawal policy on behalf of myself and my co-authors.